# Can Agricultural Socialized Services Promote the Reduction in Chemical Fertilizer? Analysis Based on the Moderating Effect of Farm Size

**DOI:** 10.3390/ijerph20032323

**Published:** 2023-01-28

**Authors:** Xiaoxuan Chen, Tongshan Liu

**Affiliations:** College of Economics and Management, Nanjing Forestry University, Nanjing 210037, China

**Keywords:** agricultural socialized services, chemical fertilizer, use intensity, farm size, the moderating effect

## Abstract

On the basis of the data of 855 farmer households in the 2020 China Land Economic Survey, this paper uses an extended regression model to empirically study the impact of agricultural socialized services on the reduction in chemical fertilizer and the moderating effect of farm size in the above impact path. The results show that adoption of agricultural socialized services by farmers can significantly promote reduction in chemical fertilizer. The moderating effect test shows that the farm size is instrumental in strengthening the effect of promoting agricultural socialized services on the reduction in chemical fertilizer. The effect of technology-intensive services on fertilizer reduction was more pronounced than that of labor-intensive services. Agricultural socialized services have a greater effect on the reduction in chemical fertilizer for farmers with a higher degree of part-time employment, but farm size can significantly enhance the fertilizer reduction effect generated by the adoption of agricultural socialized services by farmers with a lower degree of part-time employment. Therefore, we recommend further developing agricultural socialized services, strengthening the supply of agricultural green production services, and playing the role of agricultural socialized services in chemical fertilizer reduction. We also posit that encouraging large-scale farmers to adopt agricultural socialized services would further promote fertilizer reduction.

## 1. Introduction

Green agricultural development is an important way to improve the sustainable development of agriculture, and it is also an inevitable choice to enhance agricultural competitiveness [1,2]. For a long time, chemical fertilizer input has made great contributions to the increase in China’s grain production, but there are still problems related to the high application amount of chemical fertilizer and low utilization efficiency. [3,4,5]. In 2019, the amount of fertilizer used per unit area of China was 2.87 times the world average (FAOSTAT), and the fertilizer utilization efficiency of the three major food crops was 39.2% [6], which is still a small gap from the 50%~60% in developed countries in Europe and America. Long-term unreasonable use of chemical fertilizers not only endangers the quality and safety of agricultural products, but also brings great pressure to the agricultural ecological environment [7,8]. Therefore, promoting reduction in chemical fertilizers has become an inevitable requirement to promote green development of China’s agriculture. In recent years, farmers have expanded the scale of their operations through farmland transfer. Large-scale operation has played an important role in promoting fertilizer reduction and improving the efficiency of fertilizer use [9,10]. However, long-term practice shows that farmland transfer is characterized by “smallholder replication”—that is, farmland transfer is only the transfer of land cultivation rights among smallholder farmers [11], which fails to change the basic pattern of fragmented farmland management in China. In 2020, more than one-third of the country’s farmland was transferred; however, about 96% of the more than 200 million agricultural operators in China still manage farmland less than 30 mu [12]. Therefore, it is difficult to truly reach the goal of fertilizer reduction through land-scale operation only by farmland transfer [13]. How to better achieve fertilizer reduction in the context of long-term survival of small farmers in large countries has become the key to promoting green development of agriculture.

In recent years, China’s agricultural socialized services have developed rapidly. By 2020, there were 955,000 agricultural socialized service organizations nationwide, serving an area of 1.67 billion mu and more than 78 million rural households, accounting for about 37.7% of the total number of agricultural business households in the country [14]. Because of its unique advantages in alleviating the factor constraints of farmers and introducing advanced production factors, agricultural socialized services have become an important means for promoting large-scale agricultural operation and enhancing the technical efficiency of agricultural production [15,16]. Moreover, compared with ordinary farmers, agricultural socialized service subjects have more professional production knowledge, skills, and standardized production modes, which can lead farmers to change the traditional factor input modes [17,18]. Therefore, the central government began to expect agricultural socialized services to play a role in the reduction of fertilizer. The former Ministry of Agriculture issued an opinion in 2018 on vigorously implementing the rural revitalization strategy and accelerating the transformation and upgrading of agriculture, which proposed to support socialized service organizations to carry out services such as the unified distribution and application of chemical fertilizers to continuously promote chemical fertilizer reduction. The key policies to strengthen and benefit agriculture issued in 2020 by the Ministry of Agriculture and Rural Affairs and the country’s Ministry of Finance also pointed out that scientific fertilization technical services should be carried out through socialized service organizations to protect and improve the quality of cultivated land.

Many studies have also begun to focus on the role of agricultural socialized services in fertilizer reduction. Some studies pointed out a positive relationship between agricultural socialized services and fertilizer reduction [19,20]. Some scholars have also studied the causal effect between agricultural socialized services and fertilizer use. They argued that the service scale advantage of agricultural socialized services could help farmers overcome the limitations of their own factor endowment, as well as alleviate the problems of excessive fertilizer application caused by labor constraints and technological access barriers caused by financial constraints [21,22], which is helpful to reduce farmers’ fertilizer use. Biswas et al. (2021) analyzed the data of 120 rice farmers in rural areas in southwestern Bangladesh, showing that farmers who participated in agricultural extension services used less fertilizer than those who did not [23]. Huan and Zhan (2022) used the data of 1321 corn farms in China and found that the adoption of agricultural production services can significantly reduce the amount of fertilizer used in farms with the effect of machinery replacing labor and introduction of technology [24]. At the same time, as an important factor influencing the allocation of households’ resources, whether farm size will have an impact on the fertilizer reduction process of agricultural socialized services has also attracted academic attention. Using meta-analysis methods, Xie et al. (2020) showed that the promotion of fertilizer reduction by agricultural socialized services was greater among large-scale farmers compared to small-scale farmers [25]. However, Zhang et al. (2022) analyzed the data of rice farmers in Hubei Province in China and found that the fertilizer reduction effect of agricultural socialized services was more prominent among small-scale farmers [26]. Liu et al. (2022) took corn farmers in the three northeastern provinces as the research subjects, and they found that the fertilizer reduction effect of agricultural socialized services increased and then decreased with the expansion of farmland scale [27]. Of course, some scholars believed that as the service market became increasingly commercialized, agricultural socialized service organizations may collude with agricultural material dealers to make profits by inducing farmers to buy more fertilizers [28]; thus, they could not reduce farmers’ fertilizer use.

The existing literature provides a good reference for the research in this paper, but there are still some problems that need to be further discussed. First, most studies used whether to adopt agricultural socialized services as a key explanatory variable, which cannot reflect the overall level of agricultural socialized services adopted by farmers. Second, there are still few studies focusing on the differences in fertilizer reduction caused by the participation of farmers with different farm sizes in agricultural socialized services, and no consistent conclusions have been reached. Meanwhile, such research usually use group regression in the analysis of heterogeneity, but the differences in sample size brought by group regression may lead to differences in the estimated coefficients of core explanatory variables, thereby weakening the persuasiveness of the results. Therefore, this paper combined with the survey data on farmers in Jiangsu Province in China that reflect the degree of adoption of agricultural socialized services based on the situations of farmers who adopted services in six agricultural production stages, explores the impact of agricultural socialized services on fertilizer reduction. This paper also considered the interactions of agricultural socialized services with farm size, to analyze the moderating effect of farm size on the above impact path. It then further examined the differences in types of services and heterogeneity of farmers of the above influences. We expect to provide innovative ideas for promoting fertilizer reduction and agricultural socialized services.

## 2. Theoretical Analysis and Research Hypothesis

### 2.1. Agricultural Socialized Services and Fertilizer Reduction

As a professional form of production and operation, agricultural socialized services play an important role in agricultural transformation and development. Under normal circumstances, agricultural socialized services are mainly provided by service organizations that possess professional production knowledge and advanced production technology. Such agricultural socialized services with both professional and organizational composition can improve the situation of excessive fertilizer use brought about by previous empirical agriculture and the government’s difficulty in regulating farmers’ general fertilizer use behavior. Therefore, in theory, the professional effect of agricultural socialized services and the normative effect brought by its organization will have an impact on farmers’ fertilizer use.

On the one hand, agricultural socialized services can reduce the excessive use of chemical fertilizers by improving the specialization of farmers’ agricultural production and operation. Compared with most ordinary farmers who lack scientific fertilizer application knowledge and who cannot afford the high cost of agricultural machinery and production organizations developed by specialized divisions of labor, agricultural socialized service subjects not only master more knowledge of rational fertilizer use, but also have the advantage of service scale to provide a realistic way for ordinary farmers to realize agricultural mechanization [29]. By adopting agricultural socialized services, the allocation of production factors of farmers can be optimized, thereby alleviating the problem of excessive fertilizer use caused by previous empirical and manual fertilizer application. At the same time, agricultural socialized services can also introduce advanced technologies into the agricultural production process and steer farmers to apply technologies such as seedlings, deep plowing and deep loosening, soil testing, and straw returning [30,31]. These technologies can achieve reduction in chemical fertilizer use through improving the quality of seedlings, reducing fertilizer application losses, and increasing soil fertility [32,33]. In the case study of the Green Energy Company, Zhang and Luo (2019) found that the company provided fertilizer application services for farmers, including uniform soil testing and setting the nutrient elements and ratio of fertilizer in a targeted manner, which reduced the waste caused by farmers’ blind fertilizer application [34].

On the other hand, agricultural socialized services with organizational characteristics will be constrained and incentivized by the government’s standardized use of chemical fertilizer, which is conducive to the reduction in chemical fertilizer. In recent years, with the support of relevant national policies, the degree of organization of agricultural socialized services has been continuously deepened [35]. Taking agricultural machinery services as an example, from 2015 to 2020, the number of agricultural machinery service organizations in China, especially agricultural machinery cooperatives, were gradually increasing. More and more small agricultural machinery operators participated in agricultural machinery service organizations and took part in agricultural machinery operations coordinated by unified organization, which had a scale effect through organization and coordination, such as order operation, hosting operation, and cross-regional operation [36]. The government urges service subjects to standardize their own fertilizer use behavior through strict regulatory measures, such as formulating standardized service contracts, tracking and monitoring the quality of service, and establishing a blacklist system. Meanwhile, taking the needs of agricultural transformation and development into consideration, the government also regards agricultural socialized services as an important way to promote chemical fertilizer reduction [37]. A series of financial subsidies are linked with the green services provided by agricultural socialized service subjects, such as organic fertilizer application, as well as unified distribution and application of fertilizer, so as to stimulate the fertilizer reduction power of the subjects. On the basis of the above analysis, this paper proposes the following hypothesis:

**Hypothesis** **1.**
*Farmers’ adoption of agricultural socialized services can promote reduction in chemical fertilizer.*


### 2.2. The Moderating Effect of Farm Size

The above analysis mainly explored the fertilizer reduction effect of agricultural socialized services from the perspective of supply. However, from the perspective of demand subjects, farmers of different farm sizes, especially large-scale farmers, may have different needs and higher requirements in the process of adopting agricultural socialized services, which will have a new role in the fertilizer reduction effect of agricultural socialized services. Therefore, it is necessary to further investigate the role of farm size in the relationship between agricultural socialized services and farmers’ fertilizer use behavior.

From the perspective of demand, large-scale farmers who consider agriculture their industry and work hard to get rich have a stronger demand for green production services [38]. Compared with smallholder farmers, who focus more on short-term profits, large-scale farmers focus more on long-term returns, resulting in differences in technology adoption needs. Smallholder farmers tend to adopt some in-season yield technologies to increase their incomes in the short term [39]. In contrast, due to a certain scale effect, the unit cost of new technologies adopted by large-scale farmers is lower than that adopted by small-scale farmers [40], and farmers with large farms can obtain higher returns [41]; therefore, large-scale farmers have a stronger willingness and demand to adopt intertemporal technologies such as green agricultural technologies [42,43], and thus make long-term investment decisions. As Mao et al. (2021) found, households with larger farms are more likely to adopt intertemporal agricultural techniques such as straw returning than smallholder farmers [44]. However, large-scale farmers are constrained by both labor and capital, and it is difficult for them to achieve economies of scale after purchasing green production machinery; thus, large-scale farmers can overcome the conditions of insufficient green production capacity by purchasing green production services [45], thereby promoting the reduction in fertilizer use.

From the perspective of production expertise, large-scale farmers with professional production knowledge and skills will put forward higher requirements on the fertilizer use behavior of agricultural socialized service subjects. Compared with small farmers, large-scale farmers understand more information on fertilizer quality in the process of bulk purchasing of fertilizer, making it easier to distinguish high-quality fertilizer from low-quality, and they will consciously formulate scientific and reasonable fertilization plans according to their own land conditions [46,47]. Therefore, in the process of adopting agricultural socialized services, in addition to introducing more high-quality and efficient fertilizers into production, large-scale farmers will also require service subjects to apply fertilizer in combination with their own formulated fertilizer application plans. They also supervise subjects’ service behavior to reduce opportunistic tendencies and improve service efficiency. These measures make agricultural production doubly professional, helping farmers to reduce fertilizer use.

In addition, small and scattered land will increase the difficulty of agricultural machinery operation, decrease the efficiency of machinery use [48], and hinder its fertilizer reduction function. The land of large-scale farmers is relatively more contiguous and flatter, which is conducive to the operation of agricultural machinery [49], making the fertilizer application by agricultural socialized service subjects more refined and intensive, thereby reducing the amount of fertilizer use. Therefore, this paper proposes the following hypothesis:

**Hypothesis** **2.**
*Farm size has a positive moderating effect on chemical fertilizer reduction from agricultural socialized services, i.e., a larger farm size results in a greater effect of agricultural socialized services on chemical fertilizer reduction.*


## 3. Data, Variables, and Models

### 3.1. Data Source

The data used in this paper were from the China Land Economic Survey (CLES) organized by Nanjing Agricultural University in 2020. The survey adopted the sampling method of probability proportional to size (PPS). A total of 26 research districts and counties were selected from 13 prefecture-level cities in Jiangsu Province (Figure 1). Two sample townships were selected from each district and county, and one administrative village was selected from each township. Finally, 50 rural households were randomly chosen from each village. The survey covered the population of households in 2019, the agricultural production on the largest plot of contracted land, and the total production and operation of households. In the end, the survey covered a total of 2628 households in 52 administrative villages.

The plot-level survey of CLES asked how farmers plant their autumn crops on the largest plot of contracted land, including the adoption of agricultural socialized services and the corresponding fertilizer inputs. Rice and corn are the most important food crops and are also commonly grown in the survey area. There are many other crops, and the fertilizers used in different crops are quite different; hence, this paper focused on the fertilizer use intensity for rice and corn. After excluding samples that had not planted these two staple foods in the past crop year and had data gaps and outliers, 855 valid samples were finally retained.

### 3.2. Variable Description

In this paper, drawing on the research of Chang et al. (2012) [50], Zhao et al. (2021) [51] and Wang et al. (2021) [52], the fertilizer use intensity is used as the explained variable, which was measured by the average fertilizer use cost per mu. The fertilizers used by farmers included nitrogen, phosphate, potash, and compound fertilizers. Considering that there were many types of fertilizers and farmers were not sensitive to the number of fertilizers, many farmers were actually clear about the amount of fertilizer, but they were only relatively clear about the funds they had invested. Therefore, the average fertilizer cost per mu used in this paper to investigate the fertilizer use intensity of farmers can accurately reflect the situation of farmers’ fertilizer use, with a certain degree of rationality.

Agricultural socialized services are the core explanatory variable in this paper. It was measured by the degree of socialized services selected from the six stages of agricultural production: plowing, seedling, planting, pesticide spraying, harvesting and straw returning to the field. It was represented by the number of agricultural socialized services adopted by farmers, and the value was between zero and six. In order to further investigate the influence of agricultural socialized services on the fertilizer use behavior of farmers, this paper also took the cost of agricultural machinery services as a core explanatory variable for robustness testing. This variable was measured by the average machinery cost per mu for farmers’ purchases of socialized services. Generally speaking, the more agricultural socialized service stages adopted by farmers, the higher the average machinery cost per mu. Therefore, the cost of agricultural machinery services can reflect the extent of farmers’ participation in agricultural socialized services.

Farm size is the moderating variable in this paper, which was characterized by the total area of the rice and corn planted by farmers.

Farmers’ fertilizer use is influenced by a variety of factors. On the basis of existing studies [53,54], this paper selected the personal characteristics, household characteristics, and land endowment of households as the control variables. Among them, the personal characteristics of the head of the household included education; household characteristics included whether the family members participated in agricultural technical training and the proportion of household income was agricultural income; and land endowment included land fertility, irrigation conditions, and plot type for the largest parcel of land contracted by the household. The descriptive statistics of all variables are shown in Table 1.

### 3.3. Differential Analysis of Farmers’ Adoption of Agricultural Socialized Services

Table 2 shows the number of sample of farmers participating in different degrees of agricultural socialized services and their proportion in the total sample. After the samples are grouped according to the number of stages of farmers adopting agricultural socialized services, it is clear to see that samples were roughly normally distributed, indicating that most of the stages of farmers adopting agricultural socialized services were convergent. Among them, there are 70 farmers who do not adopt agricultural socialized services in the actual production process and 42 farmers who adopt them at six stages, accounting for 8.19% and 4.91% of the total sample, revealing that the proportion of farmers who do not rely on agricultural socialized services at all and absolutely relied on agricultural socialized services for production was 13.10%. This indicates that there is still immense potential for promoting agricultural socialized services in the process of rice and corn production in the future.

### 3.4. Model Specification and Estimation Methods

When analyzing the impact of agricultural socialized services on fertilizer reduction, other explanatory variables were introduced to establish the following econometric model:(1)$$Fertilizer_{i}=\alpha Service_{i}+\beta Land_{i}+\gamma X_{i}+\varepsilon_{i}$$
where $Fertilizer_{i}$ denotes the fertilizer use intensity of farmer i, $Service_{i}$ is agricultural socialized services adopted by farmer i, $Land_{i}$ refers to the total area of rice and corn planted by farmer i, $X_{i}$ represents other factors influencing the fertilizer use intensity of farmer i, including the personal characteristics of the head of household, household characteristics, and land endowment; α and β are the coefficients to be estimated, and $\varepsilon_{i}$ is the random error term.

In order to explore whether farm size had a moderating effect on the influence of agricultural socialized services on fertilizer use intensity, the interaction terms after centralized treatment between agricultural socialized services and farm size was added to model (1), and the following model was obtained:(2)$$Fertilizer_{i}=\alpha Service_{i}+\beta Land_{i}+\lambda Service_{i}\times Land_{i}+\gamma X_{i}+\varepsilon_{i}$$
where λ is the coefficient to be estimated for the interaction term, and the other variables are the same as in Equation (1).

Considering that agricultural socialized services is influenced by other factors such as head of household’s personal characteristics, household characteristics, and land endowment, it is necessary to consider the endogeneity of agricultural socialized services. Agricultural socialized services was expressed as follows:(3)$$Service_{i}=\vartheta D_{i}+\sigma_{i}$$
where $D_{i}$ is a factor influencing the agricultural socialized services, including the personal characteristics of the head of household, family characteristics, and land endowment, ϑ is the coefficient to be estimated, and $\sigma_{i}$ is a random error term.

In order to solve the estimation bias caused by the endogeneity of agricultural socialized services and considering that the fertilizer use intensity in the model is a continuous variable, this paper used the endogenous linear model in the extended regression model (ERM), which can handle endogeneity, to evaluate both Equations (1) and (3). Furthermore, the ERM framework allows endogenous variables to interact with other control variables; thus, endogenous linear models were used to estimate Equations (2) and (3). It is important to note that endogenous linear models need to contain at least one instrumental variable to be recognized. To overcome the possible endogenous problems of the model and make the econometric model identifiable, this paper drew from the idea of Ma and Abdulai (2016) in taking whether the farmers’ neighbors join cooperatives as the instrumental variable [55] and used the average number of agricultural socialized services adopted by other farmers in the same village as an instrumental variable. Due to the cohort effect, farmers’ purchases of agricultural socialized services is easily influenced by other farmers in the village. The more stages of agricultural socialized services that other farmers in the village adopt, the more stages of agricultural socialized services that the farmer will adopt. However, the number of agricultural socialized services adopted by other farmers does not directly affect the farmer’s fertilizer use.

## 4. Empirical Results and Analysis

### 4.1. The Impact of Agricultural Socialized Services and Farm Size on the Intensity of Fertilizer Use

In this paper, the influence of agricultural socialized services on the intensity of fertilizer use was first examined, and then the interaction terms between farm size and agricultural socialized services were regressed. The results are shown in Table 3. It is clear that the instrumental variable significantly affected agricultural socialized services but had no significant effect on the intensity of fertilizer use, indicating that the instrumental variable is an effective instrumental variable. Meanwhile, the residual correlation coefficients ρ between Equations (1) and (3), and between Equations (2) and (3) are significantly nonzero, and the Wald test values rejected the null hypothesis that the two equations are independent at the significance level of 1%, indicating that agricultural socialized services is endogenous, and that it is appropriate to use endogenous linear models for estimation.

The results of Equation (1) show that the estimated coefficient of agricultural socialized services is significantly negative at the 1% statistical level, indicating that agricultural socialization services have a significant inhibitory effect on the intensity of fertilizer use. In other words, under the same conditions, the adoption of agricultural socialized services can significantly reduce the intensity of fertilizer use. Hypothesis 1 is verified. In Equation (2); the interaction coefficient between agricultural socialized services and farm size is significantly negative, indicating that farm size played a positive moderating effect in the process of fertilizer reduction in agricultural socialized services. Hypothesis 2 is verified. Comparing the estimate results in Equations (1) and (2), it can be found that, after introducing interactive terms, the absolute value of the impact coefficient of agricultural socialized services on fertilizer use intensity basically increased. That is, the negative impact of agricultural socialized services on fertilizer use intensity increased, which shows that agricultural socialized services had a more significant fertilizer reduction effect on large-scale farmers.

In terms of control variables, farm size, soil fertility, and irrigation conditions significantly affect farmers’ fertilizer use intensity. The larger the farm size, the lower the fertilizer use intensity. The larger-scale farmer care more about farm income and will use fertilizer more scientifically and reasonably, accounting for cost-benefit. Soil fertility has a significant negative effect on fertilizer use intensity. When applying fertilizers, farmers mostly consider the soil conditions, and the better the soil conditions are, the lower the fertilizer input will be. Therefore, improving soil quality is also an important way to promote chemical fertilizer reduction. Irrigation conditions instead increase farmers’ fertilizer use intensity. Probably because of farmers’ widespread use of flood irrigation which can easily cause fertilizer loss, farmers need to compensate for fertilizer loss and apply more fertilizer.

### 4.2. Robustness Test

In order to test the reliability of the above estimation results, this paper replaced agricultural socialized services with the cost of agricultural machinery services for a robustness test. The results of Table 4 show that the residual correlation coefficients ρ of the farmers’ agricultural machinery service cost equation and fertilizer use intensity equation are significantly nonzero, and the Wald test values reject the null hypothesis that the two equations are independent at the given significance level. These values indicate that the average cost of agricultural machinery service and fertilizer use intensity of farmers are influenced by factors that are unobservable at the same time, and it is appropriate to use endogenous linear models for estimation.

The results of Equation (1) show that the cost of agricultural machinery service significantly reduces the intensity of fertilizer use by farmers at the 1% level; with each 1% rise in the cost of agricultural machinery service, the intensity of fertilizer use decreases by nearly 0.47 CNY/mu. The results of Equation (2) indicate that the coefficient of the interaction term between the cost of agricultural machinery service and farm size is significantly negative. Each 1% increase in the cost of agricultural machinery service signified a decrease in the intensity of fertilizer use by nearly 0.48 CNY/mu, which is more than the decrease when the interaction term is not added. The above results mean that agricultural socialized services have a stable and reliable impact on the fertilizer use intensity of farmers. A deeper participation of farmers in agricultural socialized services lead to a greater effect on fertilizer reduction, and this effect is more obvious on farmers with larger farms. Hypotheses 1 and 2 are verified again.

### 4.3. Expansive Analysis

#### 4.3.1. The Impact of Different Types of Agricultural Socialized Services on Fertilizer Reduction

Although the above study found that agricultural socialized services can promote fertilizer reduction, it did not consider the differentiated impact of services in different stages. From the perspective of factor substitution, the service of cultivation and harvesting reduces the excessive use of fertilizer caused by labor shortage through the simple replacement of labor by machine. However, the service of seeding cultivation and other stages is more technical and professional. In addition to being equipped with new mechanical equipment, these services can introduce new means of production and professional field management, which can more effectively reduce the use of fertilizer. Therefore, referring to existing research [56,57], this paper divided the six agricultural socialized services into labor-intensive services (plowing, planting, and harvesting) and technology-intensive services (seedling, pesticide spraying, and straw returning), so as to investigate the role of agricultural socialized services in different stages in fertilizer reduction.

The results of Table 5 show that both labor-intensive and technology-intensive services can significantly reduce farmers’ fertilizer use intensity. However, comparing the results in column 2 and 5, it can be seen that the absolute value of the estimated coefficient for technology-intensive services is significantly larger than that for labor-intensive services, indicating that technology-intensive services have a more negative effect on fertilizer use intensity than labor-intensive services.

The coefficients of the interaction term between farm size and both types of services are significantly negative, which indicates that farm size has a positive moderating effect in fertilizer reduction for both types of services. The change of the coefficients before and after the inclusion of the interaction term for the two types of services shows that the effect of farm size on the fertilizer reduction effect of technology-intensive services is prominent. As pointed out in the theoretical analysis, in order to improve their own agricultural operation efficiency, large-scale farmers are more willing to purchase technology-intensive services than small-scale farmers. In addition, under the consideration of cost and benefit, technology-intensive service suppliers are more inclined to provide services to large-scale farmers. Therefore, when the supply and demand sides are matched, the fertilizer reduction effect of technology-intensive services is better exhibited. This also means that, compared to achieving simply mechanized production by purchasing labor-intensive services, large-scale farmers can improve the specialization of production by purchasing technology-intensive services, which can better promote chemical fertilizer reduction.

#### 4.3.2. Heterogeneity Analysis Based on the Degree of Part-Time Employment of Farmers

Considering that there may be many differences in the adoption decisions of agricultural socialized services and the use of fertilizers among farmers with various levels of part-time employment, this paper divides the sample of farmers into two categories for heterogeneity analysis to further study the impact of agricultural socialized services on fertilizer reduction and the moderating effect of farm size. The first category was first-time farmers whose agricultural income accounted for 50–100% of the total income, while the second category was second-time farmers whose agricultural income accounted for 0–50%.

The results of group regression for both types of farmers are shown in Table 6. The results in columns 1 and 4 show that agricultural socialized services have a significant role in promoting fertilizer reduction for both first-time farmers and second-time farmers. However, compared with first-time farmers, agricultural socialized services had a greater effect on the reduction in chemical fertilizer for second-time farmers. As Sun et al. (2021) pointed out, the deeper the degree of part-time employment is, the more obvious the squeeze and income effects of non-farm employment time are, the more serious the excessive use of fertilizers by farmers is, and the stronger the demand for agricultural socialized services is [58]. Therefore, the effect of agricultural socialized services on the reduction in fertilizer of second-time farmers is higher than that of the first-time farmers.

Comparing the results of columns 3 and 5, it can be found that the interaction between agricultural socialized services and farm size is significantly negative for first-time farmers, but positive and not significant for second-time farmers, indicating that only the expansion of farm size of first-time farmers can further improve the fertilizer reduction effect of agricultural socialized services. This also means that, in order to agricultural socialized services better play a role in fertilizer reduction, it is not necessary for all farmers to expand their farm size, but to combine the production and management willingness of farmers, allowing those who truly take agriculture as their primary occupation to obtain more land management rights.

## 5. Conclusions and Policy Implications

### 5.1. Conclusions

This paper empirically analyzed the impact of agricultural socialized services on farmers’ fertilizer reduction based on the data of 855 farmers from the 2020 China Land Economic Survey, using an endogenous linear regression model. By constructing a moderating effect model, the moderating effect of farm size was tested. Moreover, this paper also examined the effects of different types of agricultural socialized services on fertilizer reduction and analyzed the heterogeneity of farmer households with different degrees of part-time employment. The results show that the adoption of agricultural socialized services by farmers can significantly contribute to fertilizer reduction. The moderating effect analysis shows that farm size can significantly enhance the effect of agricultural socialized services on fertilizer reduction. By classifying agricultural socialized services into labor-intensive and technology-intensive services, we analyzed the impact of two types of services on fertilizer reduction. It was found that the impact of technology-intensive services on fertilizer reduction was more obvious than that of labor-intensive services, and farm size had a greater impact on the fertilizer reduction of technology-intensive services. In addition, this paper also divided farmers into first-time farmers and second-time farmers according to what proportion of their income is agricultural income. The results show that the adoption of agricultural social services has a more significant fertilizer reduction effect on second-time farmers than that on first-time farmers, but farm size only significantly enhances the fertilizer reduction effect of adoption of agricultural social services by first-time farmers. This suggests that only the expansion of farm size by the low part-time farmers or farmers who truly take agriculture as their primary occupation; they can better realize the fertilizer reduction effect of agricultural social services.

### 5.2. Policy Implications

On the basis of the above research conclusions, this paper puts forward the following three policy implications: first, governments should attach importance to the role of agricultural socialized services in promoting reduction in chemical fertilizer. Relevant departments should vigorously cultivate and improve the agricultural socialized service market. They should not only provide agricultural socialized service subjects with financial subsidies and projects to attract more market entities to participate in the service supply, but also actively encourage and guide more farmers to be involved in the market. Second, governments should focus on increasing policy support for agricultural socialized service subjects in the purchase of green production machinery and the research and application of green production technology. These actions can help service subjects expand the type of supply and improve service quality, so as to better play the role of socialized services in green agricultural production. Third, it is necessary to continue to improve the land transfer market while cultivating the agricultural socialized service market. Local governments need to create conditions for more farmers who consider agriculture their primary occupation to expand their farm size, so as to realize the synergistic effect between agricultural moderate-scale operation and agricultural socialized services in promoting agricultural green production.

There were also some potential limitations in this paper. On the one hand, the data used in this paper were cross-sectional, which may have brought about some endogenous problems caused by unobservable variables that do not change over time. On the other hand, the research area of this paper was Jiangsu Province and did not cover most areas of China; thus, it was impossible to explore the differences in fertilizer reduction caused by agricultural socialized services in different regions. In subsequent studies, we will further expand the temporal and spatial scales of the study in order to obtain more general results.

## Figures and Tables

**Figure 1 ijerph-20-02323-f001:**
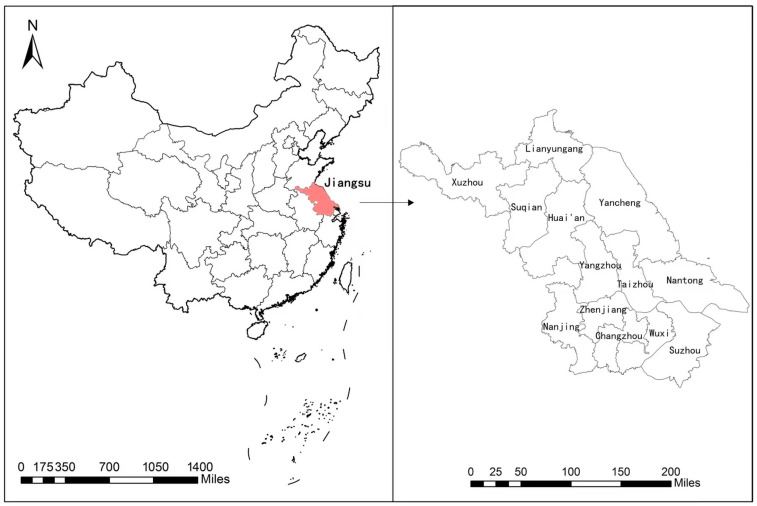
Distribution map of the sample area.

**Table 1 ijerph-20-02323-t001:** Variable definition and descriptive statistics.

Variable	Definition	Mean	SD
Fertilizer use intensity	Total cost of fertilizer input by farmers planting two kinds of grains (CNY/mu)	192.487	70.706
Agricultural socialized services	The number of socialized services adopted by farmers, ranging from 0 to 6	3.028	1.509
The cost of agricultural machinery services	Logarithm of average mechanical cost per mu of farmers purchasing socialized services (CNY/mu)	4.569	1.531
Labor-intensive services	The number of services adopted by farmers in ploughing, planting and harvesting	1.960	0.944
Technology-intensive services	The number of services adopted by farmers in seedling, pesticide spraying and straw returning	1.068	0.775
Farm size	The total area of two kinds of grains planted by farmers(mu)	19.494	62.850
Education	Actual years of education of the head of household (years)	6.890	3.552
Agricultural technology training	Whether family members have attended agricultural technical training in the past year: 1 = yes, 0 = no	0.353	0.478
Proportion of agricultural income	Percentage of agricultural income in total household income	0.261	0.307
Soil fertility			
Poor	1 = poor, 0 = medium and good	0.088	0.284
Medium	1 = medium, 0 = poor and good	0.500	0.500
Good	1 = good, 0 = poor and medium	0.412	0.492
Irrigation conditions	Whether the land can be irrigated:1 = yes, 0 = no	0.905	0.294
Plot type			
depression	1 = depression, 0 = flat land and sloping land	0.035	0.184
flat ground	1 = flat land, 0 = depression and sloping land	0.935	0.247
sloping land	1 = sloping land, 0 = depression and flat land	0.030	0.172
IV	The average number of agricultural socialized services adopted by other farmers in the same village	3.017	0.679

**Table 2 ijerph-20-02323-t002:** The number and proportion of agricultural socialized services adopted by farmers with different numbers of stages.

Index	The Number of Stages of Farmers Adopting Agricultural Socialized Services						
	0	1	2	3	4	5	6
The number of sample	70	45	170	263	165	100	42
Proportion of the total sample (%)	8.19	5.26	19.88	30.76	19.30	11.70	4.91

**Table 3 ijerph-20-02323-t003:** Estimated results of the impact of agricultural socialized services on fertilizer use intensity: based on the moderating effect of farm size.

Variable	(3)	(1)	(3)	(2)
Agricultural socialized services	—	−24.444 *** (5.514)	—	−25.816 *** (5.619)
Agricultural socialized services × farm size	—	—	—	−0.040 ** (0.018)
Farm size	−0.001 (0.001)	−0.088 * (0.045)	−0.001 (0.001)	−0.110 ** (0.054)
Education	−0.006 (0.014)	0.039 (0.724)	−0.006 (0.014)	0.061 (0.728)
Agricultural technology training	−0.099 (0.107)	−7.745 (5.942)	−0.099 (0.107)	−8.639 (6.000)
Proportion of agricultural income	−0.115 (0.168)	11.789 (9.545)	−0.115 (0.168)	11.709 (9.664)
Soil fertility(poor as the control group)	—	—	—	—
Medium	−0.226 (0.181)	−24.978 ** (12.211)	−0.226 (0.181)	−25.069 ** (12.270)
Good	−0.483 *** (0.186)	−29.445 ** (13.064)	−0.483 *** (0.186)	−29.759 ** (13.119)
Irrigation conditions	0.152 (0.180)	14.555 * (8.112)	0.152 (0.180)	13.239 (8.306)
Plot type(depression as the control group)	—	—	—	—
flat ground	−0.175 (0.224)	10.494 (12.328)	−0.175 (0.224)	10.683 (12.415)
sloping land	−0.598 * (0.334)	−2.048 (17.229)	−0.598 * (0.334)	−2.776 (17.428)
IV	0.756 *** (0.079)	—	0.756 *** (0.079)	—
Constant	1.236 *** (0.437)	270.221 *** (30.097)	1.236 *** (0.437)	275.808 *** (30.515)
Wald chi2	30.63 ***		34.26 ***	
ρ	0.410 *** (0.088)		0.431 *** (0.088)	
N	855			

Note: ***, **, * denote significance at 1%, 5%, and 10% level, respectively; robust standard errors are presented in parentheses.

**Table 4 ijerph-20-02323-t004:** Robustness test based on the cost of agricultural machinery services.

Variable	(3)	(1)	(3)	(2)
The cost of agricultural machinery services	—	−47.135 *** (14.650)	—	−47.987 *** (14.733)
The cost of agricultural machinery services × farm size	—	—	—	−0.028 * (0.014)
Farm size	−0.003 ** (0.001)	−0.203 ** (0.085)	−0.003 ** (0.001)	−0.231 ** (0.096)
Education	−0.016 (0.015)	−0.556 (0.985)	−0.016 (0.015)	−0.550 (0.99)
Agricultural technology training	0.022 (0.113)	−4.295 (7.364)	0.022 (0.113)	−4.634 (7.432)
Proportion of agricultural income	−0.078 (0.184)	10.928 (12.553)	−0.078 (0.184)	10.981 (12.71)
Soil fertility(poor as the control group)	—	—	—	—
Medium	−0.259 * (0.154)	−31.682 ** (14.635)	−0.259 * (0.154)	−31.19 ** (14.700)
Good	−0.359 ** (0.158)	−34.553 ** (15.799)	−0.359 ** (0.158)	−34.388 ** (15.873)
Irrigation conditions	0.627 *** (0.210)	40.391 *** (13.079)	0.627*** (0.210)	40.053 *** (13.151)
Plot type(depression as the control group)	—	—	—	—
flat ground	−0.111 (0.231)	9.538 (16.158)	−0.111 (0.231)	9.013 (16.155)
sloping land	−0.262 (0.383)	0.230 (22.883)	−0.262 (0.383)	−0.854 (23.107)
IV	0.392 *** (0.076)	—	0.392 *** (0.076)	—
Constant	3.39 *** (0.426)	399.817 *** (76.950)	3.39 *** (0.426)	404.419 *** (77.341)
Wald chi2	17.60 *		20.90 **	
ρ	0.700 *** (0.111)		0.707 *** (0.108)	
N	855			

Note: ***, **, * indicate significance at 1%, 5% and 10% levels, respectively, with robust standard error in brackets.

**Table 5 ijerph-20-02323-t005:** The impact of different types of agricultural socialized services on fertilizer reduction.

Variable	(1)	(2)	Variable	(1)	(2)
Labor-intensive services	−33.181 *** (8.944)	−35.154 *** (9.116)	Technology-intensive services	−63.014 *** (15.219)	-65.639 *** (15.440)
Labor-intensive services × farm size	—	−0.061 ** (0.029)	Technology-intensive services × farm size	—	−0.082 ** (0.040)
Control variables	YES	YES	Control variables	YES	YES
Constant	255.891 *** (30.426)	267.689 *** (32.946)	Constant	266.459 *** (30.644)	269.942 *** (31.149)
Wald chi2	22.79 **	25.92 ***	Wald chi2	28.25 ***	31.40 ***
ρ	0.349 *** (0.098)	0.368 *** (0.097)	ρ	0.535 *** (0.103)	0.550 *** (0.101)
N	855		N	855	

Note: *** and ** indicate significance at 1% and 5% levels, respectively; in parentheses are robust standard errors; due to limited space, the regression results of the control variables and Equation (3) are not reported, the same below.

**Table 6 ijerph-20-02323-t006:** Heterogeneity analysis based on the degree of part-time employment of farmers.

Variable	The First-Time Farmers		The Second-Time Farmers	
	(1)	(2)	(1)	(2)
Agricultural socialized services	−17.276 ** (8.660)	−19.704 ** (8.942)	−20.286 *** (5.668)	−19.945 *** (6.451)
Agricultural socialized services × farm size	—	−0.053 ** (0.022)	—	0.013 (0.117)
Control variables	YES	YES	YES	YES
Constant	214.236 *** (57.015)	229.017 *** (58.821)	277.887 *** (30.591)	276.824 *** (32.053)
Wald chi2	20.77 **	25.64 ***	26.38 ***	26.41 ***
ρ	0.263 (0.176)	0.377 ** (0.168)	0.388 *** (0.096)	0.388 *** (0.096)
N	194		569	

Note: ***, ** denote significance at 1% and 5% levels, respectively; In parentheses are robust standard errors; since it excludes isolated farmers whose agricultural income is zero and pure farmers whose agricultural income accounts for 100%, the total sample is less than 855.

## Data Availability

The data underlying the results presented in the study are all available. The data presented in this study are available on request from the corresponding author.

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
