# Peer review of "Can Agricultural Socialized Services Promote the Reduction in Chemical Fertilizer? Analysis Based on the Moderating Effect of Farm Size"

_ijerph, 2023, doi:10.3390/ijerph20032323_

Round 1
Reviewer 1 Report
Language needs to be polished
As a key variable, fertilizer use intensity is not a useful indicator. It is important to note that a high level of fertilizer use intensity does not necessarily mean that farmers applied a greater amount of fertilizer, since some organic fertilizers are more expensive than chemical fertilizers. The prices of different bands of the chemical company may also differ significantly.
The authors considered only the size of the farm but did not take into account the number of plots of land. In the case of a large scale farm with a large number of land plots, the results regarding the size of the farm will be biased.
In table 3, the result of the number of stages that farmers adopt agricultural socialized services (0 as the control group) does not seem convincing, it suggests that if a household adopts 6 stages of agricultural socialized services, fertilizer use intensity will decrease by 180.332? It does not seem to make sense.
According to Table 1, the data of the cost of agricultural machinery services appears to be reported incorrectly. If the national logarithm of this variable is 162, the original values would be much higher than 1 billion. In addition, 91.8% of household heads are male, so I believe the gender variable could be omitted.
In fact, I am confused by the first row of table 2-7. What do you mean by (3), (2), (1), (3)?
Do the terms "circulation of farmland" and "land transfer" mean the same thing? It is only appropriate to use the term "land transfer" since farmland is not a currency or newspaper that can be circular.
Line 21, you cannot say that the government should act. The policy also does not guarantee that it will function as you suggest, since you are not 100% certain it will work as intended.
Please provide more detail for the words "other agricultural chemicals".
Line 33-34, this data does not provide any further information as to whether China had been overusing chemical fertilizers. Overuse of fertilizer can only be compared with your own condition. Some regions in the world, such as South-Africa, have bad soil conditions, and they do not have the funds or sources to buy or apply chemical fertilizers, so chemical fertilizers are underutilized.
Line 42, What do you mean by “smallholder replication ”?
Add data sources to line 44 and update the data.
Line 52, add data sources.
Line 62-67, this opinion did not mention that "the unified distribution and application of chemical fertilizers"
Author Response
Please refer to the file in the attachment.

Reviewer 2 Report
Please check the attachment.

Reviewer 3 Report
Comments to the Author:
Overall, the manuscript’s analysis is interesting in Agricultural Socialized Services and Chemical Fertilizer. However,The authors should revise the empirical content of the article more extensively. The following comments may be useful for future submissions.
Problems and Recommendations
1. In line 254, there are only five stages? If the core explanatory variables are defined as ordered variables, the author should emphasise the serial relationships represented by the values in the article.
2.The authors explain the method used when equation 3 was estimated, linear or ordered regression.
3. The authors estimated the coefficients for each of the six stages, then the interaction term should contain all ordered variables when estimating equation 2.
4. The meaning of ρ in the table should be introduced in the text.
5. The above questions also apply to Tables 6 and 7.
6. In Table 4, the marginal effects are interpreted as to whether fertilizer use intensity is justified
Other Questions
7. in line 42, "smallholder replication" is not understood
8. The author needs to indicate the source of the data, lines 44-46 and 52-54
9. in line 62, "former" is not understood
10.184-187 lines, whether supported by literature
11. The authors should provide a brief description of the estimated coefficients for the control variables in Table 3.
12. The variables in Tables 5 and 7 should be shown in the statistical description
